# Four New Species of *Aspergillus* Subgenus *Nidulantes* from China

**DOI:** 10.3390/jof8111205

**Published:** 2022-11-15

**Authors:** Bingda Sun, Chunling Luo, Gerald F. Bills, Jibing Li, Panpan Huang, Lin Wang, Xianzhi Jiang, Amanda Juan Chen

**Affiliations:** 1China General Microbiological Culture Collection Centre, Institute of Microbiology, Chinese Academy of Sciences, Beijing 100101, China; 2State Key Laboratory of Organic Geochemistry and Guangdong-Hong Kong-Macao Joint Laboratory for Environmental Pollution and Control, Guangzhou Institute of Geochemistry, Chinese Academy of Sciences, Guangzhou 510640, China; 3Texas Therapeutics Institute, The Brown Foundation Institute of Molecular Medicine, University of Texas Health Science Center at Houston, Houston, TX 77054, USA; 4Microbiome Research Center, Moon (Guangzhou) Biotech Ltd., Guangzhou 510535, China

**Keywords:** Ascomycota, Eurotiales, multigene phylogeny, macromorphology

## Abstract

*Aspergillus* subgenus *Nidulantes* includes species with emericella-like ascomata and asexual species. Subgenus *Nidulantes* is the second largest subgenus of *Aspergillus* and consists of nine sections. In this study, agricultural soils were sampled from 12 provinces and autonomous regions in China. Based on primary BLAST analyses, seven of 445 *Aspergillus* isolates showed low similarity with existing species. A polyphasic investigation, including phylogenetic analysis of partial ITS, β-tubulin, calmodulin, and RNA polymerase II second largest subunit genes, provided evidence that these isolates were distributed among four new species (*Aspergillus guangdongensis*, *A. guangxiensis*, *A. sichuanensis* and *A. tibetensis*) in sections *Aenei*, *Ochraceorosei,* and *Sparsi* of subgenus *Nidulantes*. Illustrated morphological descriptions are provided for each new taxon.

## 1. Introduction

*Aspergillus* is a monophyletic genus comprising six subgenera: subgenus *Aspergillus*, *Cremei*, *Circumdati*, *Fumigati*, *Nidulantes*, *Polypaecilum* [1,2,3]. Subgenus *Nidulantes* (type species: *Aspergillus nidulans*) is the second largest subgenus after *Circumdati* and includes over 100 species [4]. However, only a few species have been reported from China. Most of species in this subgenus exhibit biseriate conidial heads, brown-pigmented conidiophores, globose and echinulate conidia. For some species (mostly in sections *Aenei*, *Nidulantes* and *Usti*), sexual states with reddish brown, violet or dark brown ascomata embedded in masses of Hülle cells are produced. Sexual species previously were named *Emericella* under the dual nomenclature system [5]. Ascospores produced by species in subgenus *Nidulantes* often bear equatorial ridges, reticulations and appendages that are morphologically distinct from the sexual states of other subgenera in *Aspergillus* [4,6,7,8,9,10,11,12,13,14,15,16,17,18,19,20,21,22,23,24].

Initially, five sections were established in subgenus *Nidulantes* based on morphological features: *Nidulantes*, *Versicolores*, *Usti*, *Terrei*, and *Flavipedes* [25]. Subsequently, DNA sequences of the β-tubulin (*BenA*), calmodulin (*CaM)*, RNA polymerase II genes (*RPB2*), and the ITS, large subunit rDNA sequences (LSU) were analyzed in combination to determine infrageneric relationships in *Aspergillus*. Sections *Nidulantes*, *Sparsi*, *Ochraceorosei,* and *Usti* formed monophyletic groups along with other outlying species, e.g., *A. raperi*, *A. ivoriensis*, *A. silvaticus,* and *A. bisporus* [26,27]. Section *Aenei* was later introduced based on phylogenetic analyses and included additional species [28]. Most recently, application of a comprehensive polyphasic approach defined nine sections in the subgenus, leading to the addition of section *Cavernicolarum* (as ‘*Cavernicola*’) [4]. Later, *A. pepii* was described in section *Versicolores* [29]; *A. amethystinus*, *A. askiburgiensis*, *A. coloradensis*, *A. croceus*, *A. croceiaffinis*, *A. dipodomyus*, *A. longistipitatus*, *A. stelliformis,* and *A. tumidus* were described in section *Nidulantes* [24,30,31]; *A. contaminans* and *A. sigarelli* were described in section *Usti* [32,33]. Houbraken et al. gave an overview of families and genera of the Eurotiales and introduced an updated subgeneric, sectional and series classification [34]. Recently, a broad species concept with only four species, namely *A. versicolor*, *A. creber*, *A. sydowii,* and *A. subversicolor,* was introduced in *Aspergillus* series *Versicolores* [35].

Temperature growth profiles are useful characters for distinguishing species in subgenus *Nidulantes* [4]. In section *Nidulantes*, most of the sexual species with globose ascospores grow optimally at 37 °C, while most of asexual and sexual species with stellate ascospores grow optimally at 27 to 37 °C [4]. In section *Aenei*, all species fail to grow at or above 40 °C [28]. Ascomata and ascospores are generally produced by members in sections *Nidulantes* and *Aenei*. Hülle cells that surround ascomata are specialized thick-walled cells associated with subgenus *Nidulantes*. In some species, emericella-like ascomata may be absent, yet the mycelium may give rise to masses or embedded Hülle cells, e.g., species in sections *Aenei*, *Usti*, *Sparsi*, and occasionally in others [4,17,28,36,37,38].

Species of subgenus *Nidulantes* are widespread, although they tend to be more common in warm, arid, and saline environments [15,17,19,21,39,40]. A few species in subgenus *Nidulantes* have been reported to cause human infections, including *A. nidulans*, *A. quadrilineatus*, *A. sublatus*, and *A. delacroxii* (=*A. spinulosporus*), all of which grow well at 37 °C and have been reported as etiologic agents of aspergillosis [41,42]. Other species, such as *A. protuberus*, *A. unguis*, *A. sydowii*, and *A. hongkongensis*, which grow poorly or do not grow at 37 °C were mostly responsible for superficial infections, such as onychomycosis, keratitis, and scalp mycosis [42,43,44,45,46,47,48,49].

*Aspergillus nidulans* is the model fungus for understanding eukaryotic cell biology and molecular processes [50]. Its whole genome was sequenced in 2005 [51] to study a wide range of cellular attributes, including recombination, genome editing, DNA repair, mutation, cell cycle control, nucleokinesis, pathogenesis, secondary metabolism, and experimental evolution [52,53,54,55,56,57,58]. In addition, species of subgenus *Nidulantes* are well known for their complex secondary metabolism. *Aspergillus nidulans* FGSC A4 is arguably the most intensively studied fungal strain with regard to its secondary metabolome [59]. *Aspergillus pachycristatus* is the commercial source of echinocandin B, the natural product precursor for the antifungal drug anidulafungin [60] and the clinical candidate rezafungin [61].

In this study, soils were sampled from 11 provinces (Anhui, Guangdong, Guangxi, Guizhou, Hainan, Henan, Jiangsu, Shandong, Shanxi, Sichuan, Yunnan) and the Tibet autonomous region in China during 2017. After selecting strains from soil isolation plates, all isolates were sequenced, and sequences were interrogated by public database searches. Based on sequence analysis, seven isolates (CGMCC 3.19704, 3.19705, 3.19706, 3.19707, 3.19708, 3.19709, 3.19710) showed low similarity to previously sequenced species, and further phylogenetic analysis placed these strains in subgenus *Nidulantes*. A multi-gene phylogeny was constructed from the ITS, *BenA*, *CaM,* and *RPB2* gene sequences. Morphological features have been used to corroborate their phylogenetic distinctions and to define and describe four new species.

## 2. Materials and Methods

### 2.1. Soil Collection

Soils were sampledfrom farmlands and forests from 11 provinces in China, including Anhui, Guangdong, Guangxi, Guizhou, Hainan, Henan, Jiangsu, Shandong, Shanxi, Sichuan, Yunnan, and the Tibet autonomous region during 2017. At each sampling site, the superficial soil layer (approximately top 5 cm) was removed, and deeper soil was carefully excavated and sealed in a sterile plastic bag. Longitude and latitude of sampling locations and vegetation information were recorded. Soil samples were transported to the lab within three days.

### 2.2. Fungal Isolation and Preliminary Identification

From each soil collection, approximately 10 g were sieved to remove large particles, rocks, and debris, and then suspended in 90 mL sterile water. Aliquots of 10*^−^*^3^ dilutions were spread onto potato dextrose agar (PDA, Guangdong Huankai Microbiological Technology Co., Ltd. (Guangzhou, China)) and rose bengal medium (RBM, Beijing Luqiao Technology Co., Ltd. (Beijing, China)) with tetracycline hydrochloride and chloramphenicol at 100 μg/mL added after autoclaving. After three to five days, emerging colonies were removed and transferred to new PDA plates. After 1 to 2 weeks of growth, strains were sorted into approximate morphological types and identified further by microscopy. Strains belonging to the genus *Aspergillus* were submitted for DNA extraction using the Ultraclean^TM^ Microbial DNA isolation Kit (MoBio Laboratories Inc., Solana Beach, CA, USA) and CaM gene sequencing for accurate identification. The living ex-type strains of described new species were deposited in China General Microbiological Culture Collection Centre, Institute of Microbiology, Chinese Academy of Sciences (CGMCC, https://cgmcc.net/english/ accessed on 1 July 2022) and dried cultures were deposited in herbarium (HMAS) located in same institute (accessed on 1st July 2022).

### 2.3. Phylogenetic Analysis

For strains (Table 1) unassignable to a known species, *BenA*, *CaM*, and *RPB2* gene fragments were amplified using previously described primers and programs [62] and sequenced via ABI 3730XL DNA sequencer (Applied Biosystems). Reference sequences from strains of subgenus *Nidulantes* were used for phylogenetic reconstruction and placement of unknown isolates [4,24,29,31,32,33]. Aligned sequences were analyzed with FindModel (http://hiv.lanl.gov/content/sequence/findmodel/findmodel.html, accessed on 1 January 2022) to select the most appropriate model of nucleotide substitution [63]. Maximum likelihood analyses with 1000 bootstrap replicates were run using RAxML [64]. Bayesian analyses were run with MrBayes v. 3.2 [65]. The sample frequency was set to 100, and the first 25% of trees were removed as a burn-in. *BenA*, *CaM* and *RPB2* sequences from *A. flavipes* NRRL 302 were used as outgroup. The resulting trees were analyzed with FigTree v1.4.2 (http://tree.bio.ed.ac.uk/software/figtree/, accessed on 1 January 2022) and visualized using Adobe Illustrator CS5. BI posterior probabilities (pp) values and bootstrap (bs) percentages of analysis were labeled at the branch nodes. Values less than 0.95 pp and less than 70% bs were not shown. Branches with values more than 1.00 pp and 95% bs are thickened.

### 2.4. Morphology

Macroscopic and microscopic characters were observed and measured as previously described. The isolates were inoculated and incubated for 7 d on the agar media Czapek yeast autolysate agar (CYA), yeast extract sucrose agar (YES), creatine sucrose agar (CREA), dichloran 18% glycerol agar (DG18), oatmeal agar (OA), and malt extract agar (MEA; Oxoid CM0059) [4]. Colony colors were referenced to numbered color codes in parentheses [66]. Light microscope preparations were made from one wk old colonies grown on MEA. To induce ascomata formation, colonies were incubated for more than two weeks. A Zeiss Stereo Discovery V20 dissecting microscope and a Zeiss AX10 Imager A2 light microscope, both equipped with an Axiocam 506 color camera and ZEN v.2.0 software (made in Germany), were used to capture digital images.

## 3. Results

### 3.1. Phylogeny

We used concatenated sequence data combining the ITS, *BenA*, *CaM* and *RPB2* regions for defining relationships within subgenus *Nidulantes*. The total length of the aligned data set was 2772 characters, containing 626, 576, 651, and 919 bp for ITS, *BenA*, *CaM*, and *RPB2* respectively. K2P + G model was used for ITS, GTR+G model was used for *BenA*, *CaM* and *RPB2*. The alignment sequences and the tree file were deposited in TreeBASE (S29781, https://www.treebase.org accessed on 14 October 2022).

Based on multi-gene phylogeny, subgenus *Nidulantes* was classified into nine sections (*Aenei*, *Bispori*, *Cavernicolarum*, *Nidulantes*, *Ochraceorosei*, *Raperorum*, *Silvatici*, *Sparsi* and *Usti*). The four new species were resolved into sections *Aenei*, *Ochraceorosei,* and *Sparsi*, respectively (Figure 1).

Section Aenei (1.00 pp, 100 bs) includes *A. aeneus*, *A. bicolor*, *A. crustosus*, *A. discophorus*, *A. eburneocremeus*, *A. foeniculicola*, *A. heyangensis*, *A. karnatakaensis*, *A. spectabilis,* and two new species, *A. sichuanensis* and *A. tibetensis*. *Aspergillus sichuanensis* is represented by three strains isolated from soils under pea cultivation, Sichuan province, China. They formed a strongly supported clade closely related with *A. heyangensis* and *A. crustosus* in the multi-gene phylogeny and the BenA, CaM and RPB2 single-gene trees (Figure 1 and Appendix A). In the ITS phylogeny, *A. sichuanensis* clusters outside the clade containing *A. crustosus*, *A. heyangensis* and *A. tibetensis* (Appendix A). *Aspergillus tibetensis* clustered outside the groups with *A. crustotus*, *A. heyanensis* and *A. sichuanensis* in the multi-gene phylogeny and in the BenA, CaM and RPB2 single-gene trees (Figure 1 and Appendix A). In the ITS phylogeny, *A. sichuanensis* formed a statistically unsupported clade with *A. crustosus* and *A. heyanensis* (Appendix A).

Section *Sparsi* (1 pp, 100 bs) includes *A. amazonicus*, *A. anthodesmis*, *A. biplanus*, *A. conjunctus*, *A. diversus*, *A. haitiensis*, *A. implicatus*, *A. panamensis*, *A. sparsus*, and the new species *A. guangxiensis*. This species formed a highly supported clade with *A. amazonicus*, *A. anthodesmis*, *A. conjunctus,* and *A. panamensis* in the multi-gene phylogeny and the *CaM* and *RPB2* single-gene trees (Appendix A). In the ITS and *BenA* phylogenies, *A. biplanus* and *A. diversus* were also included in this clade, but the statistical support for this clade was low. The last new species, *A. guangdongensis,* fell in section *Ochraceorosei* and is closely related to *A. funiculosus* in the multi-gene phylogeny and all single-gene trees (Figure 1 and Appendix A).

### 3.2. Taxonomy

Aspergillus guangdongensis B.D. Sun, X.Z. Jiang & A.J. Chen, sp. nov. Figure 2.

MycoBank: MB837898.

Diagnosis: *Aspergillus guangdongensis* produces uniseriate sterigmata, globose vesicles measuring 25–35 μm, ellipsoidal, smooth conidia, orange in mass.

Typification: CHINA, Guangdong, isolated from farmland soil under bean cultivation, 2017, P.P. Huang (holotype HMAS 248373). Ex-type culture CGMCC3.19704 = MN014767. GenBank: ITS = MN640760; *BenA* = MN635246; *CaM* = MN635257; *RPB2* = MN635269.

Etymology: Latin, name refers to its origin, isolated from Guangdong province, China.

Description: Colony diam—7 d (mm): CYA 30–33; CYA 37 °C Microcolonies 3–4; MEA 43–44; OA 43–44; YES 51–54; CREA 23–24; DG18 28–29. Colony characters—CYA 25 °C, 7 d: Colonies deep, plane; margins entire; mycelium sulphur yellow (15); texture floccose; sporulation sparse; conidia en masse greyish yellow-green (68); soluble pigments absent; exudates absent; reverse rust (39) and orange (7). MEA 25 °C, 7 d: Colonies moderately deep, plane; margins entire; mycelium greyish yellow-green (68); texture floccose; sporulation dense; conidia en masse primrose (66); soluble pigments absent; exudates absent; reverse primrose (66). YES 25 °C, 7 d: Colonies deep, sulcate; margins entire; mycelium sulphur yellow (15); texture floccose; sporulation sparse; conidia en masse greyish yellow-green (68); soluble pigments absent; exudates absent; reverse rust (39) and orange (7). DG18 25 °C, 7 d: Colonies moderately deep, plane; margins entire; mycelium sulphur yellow (15); texture floccose; sporulation sparse; conidia en masse greyish yellow-green (68); soluble pigments fawn (87); exudates absent; reverse orange (7) at middle, pale luteous (11) at edge. OA 25 °C, 7 d: Colonies moderately deep, plane; margins entire; mycelium sulphur yellow (15); texture floccose, crust-like in middle; sporulation sparse; conidia en masse greyish yellow-green (68); soluble pigments fawn (87); exudates clear droplets; reverse saffron (10). CREA 25 °C, 7 d: Moderate growth, acid production absent.

Ascomata and Hülle cells not observed. Conidiophores with smooth stipes, light brown, 230–450 × 9–11.5 μm; vesicles light brown, globose, 25–35 μm wide, fertile over the two thirds, uniseriate, phialides hyaline, flask-shaped, 7.5–8 × 2.5–3 μm. Conidia smooth, orange in mass, subglobose to ellipsoidal, 3–4 × 2–3 μm.

Notes: Morphologically and phylogenetically, *A. guangdongensis* is closely related to *A. funiculosus* (Table 2). Both species produce yellow colonies, uniseriate and globose vesicles, however *A. funiculosus* produces two kinds of conidia: smooth, elliptical, faintly colored conidia and coarsely rugulose, globose, yellow-brown conidia [17]. According to BLAST analyses, *A. guangdongensis* is differ from *A. funiculosus* in ITS (96% similarity), *BenA* (93% similarity), *CaM* (91% similarity) and *RPB2* (95% similarity) sequences.

*Aspergillus guangxiensis* B.D. Sun, X.Z. Jiang & A.J. Chen, sp. nov. Figure 3.

MycoBank: MB837899.

Diagnosis: *Aspergillus guangxiensis* grows restrictedly, produces globose vesicles measuring 17–28 μm, produces subglobose to globose, finely roughened conidia measuring 3–4 μm.

Typification: CHINA, Guangxi, isolated from farmland soil under maize cultivation, 2017, P.P. Huang (holotype HMAS 248372). Ex-type culture CGMCC3.19709 = MN115635. GenBank: ITS = MN640765; *BenA* = MN635251; *CaM* = MN635262; *RPB2* = MN635274.

Etymology: Latin, name refers to its origin, isolated from Guangxi province, China.

Description: Colony diam—7 d (mm): CYA 11–13; CYA 37 °C No growth; MEA 9–11; OA 13–14; YES 12–14; CREA 6–8; DG18 4–5. Colony characters—CYA 25 °C, 7 d: Colonies moderately deep, plane; margins entire; mycelium buff (45); texture floccose; sporulation dense; conidia en masse greyish yellow-green (68); soluble pigments absent; exudates absent; reverse buff (45). MEA 25 °C, 7 d: Colonies moderately deep, plane; margins entire; mycelium white; texture floccose; sporulation dense; conidia en masse greyish yellow-green (68); soluble pigments absent; exudates absent; reverse buff (45). YES 25 °C, 7 d: Colonies moderately deep, plane; margins entire; mycelium white; texture floccose; sporulation dense; conidia en masse greyish glaucous (91); soluble pigments absent; exudates absent; reverse pale luteous (11). DG18 25 °C, 7 d: Colonies moderately deep, plane; margins entire; mycelium white; texture floccose; sporulation absent; soluble pigments absent; exudates absent; reverse orange (7). OA 25 °C, 7 d: Colonies moderately deep, plane; margins entire; mycelium white and straw (46); texture floccose; sporulation moderately dense; conidia en masse dark green (21); soluble pigments absent; exudates clear droplets; reverse pure yellow (14) CREA 25 °C, 7 d: Acid production absent.

Ascomata and Hülle cells not observed. Conidiophores with smooth stipes, light brown, 350–600 × 5.5–10 μm; vesicles hyaline, globose, 17–28 μm wide, fertile over the entire surface, metulae hyaline, 6.5–7.5 × 4–6 μm; phialides hyaline, flask-shaped, 7–9 × 2.5–3.5 μm. Conidia finely roughened, pale green in mass, subglobose to globose, 3–4 μm.

Notes: In the phylogenetic analyses, the two *A. guangxiensis* strains were positioned at a unique branch basal to branches with *A. conjunctus*, *A. panamensis* and *A. anthodesmis* (Figure 1).

Morphologically, *A. guangxiensis* resembles *A. conjunctus* and *A. haitiensis* because all three species produce globose vesicles measuring around 15 to 30 μm, however, *A. conjunctus* grows faster, reaching 30 mm in 10 days on CA [17], while *A. haitiensis* produces larger globose to ellipsoidal smooth conidia measuring 4–5.6 × 5–6 μm [38]. According to BLAST analyses, ITS of *A. guangxiensis* is close to *A. haitiensis* and *A. sparsus* (92% similarity), *BenA* sequence is close to *A. amazonicus* and *A. conjunctus* (88% similarity), *CaM* sequence is close to *A. anthodesmis* (87% similarity), *RPB2* sequence is close to *A. heyangensis* (94% similarity).

*Aspergillus sichuanensis* B.D. Sun, X.Z. Jiang & A.J. Chen, sp. nov. Figure 4.

MycoBank: MB837900.

Diagnosis: *Aspergillus sichuanensis* produces Hülle cells, long conidiophores measuring 200–500 μm and echinulate green conidia measuring 3–4 μm.

Typification: CHINA, Sichuan, isolated from farmland soil under pea cultivation, 2017, P.P. Huang (holotype HMAS 248374). Ex-type culture CGMCC3.19706 = MN18437. GenBank: ITS = MN640762; *BenA* = MN635248; *CaM* = MN635259; *RPB2* = MN635271.

Etymology: Latin, name refers to its origin, isolated from Sichuan province, China.

Description: Colony diam—7 d (mm): CYA 14–23; CYA 37 °C 5–6; MEA 13–16; OA 14–20; YES 18–28; CREA 9–20; DG18 13–18. Colony characters—CYA 25 °C, 7 d: Colonies moderately deep, sulcate; margins slightly irregular; mycelium white; texture floccose; sporulation sparse; conidia en masse dark green (21); soluble pigments absent; exudates absent; reverse primrose (66). MEA 25 °C, 7 d: Colonies moderately deep, plane; margins slightly irregular; mycelium white; texture floccose; sporulation moderately dense; conidia en masse dark green (21); soluble pigments absent; exudates absent; reverse buff (45). YES 25 °C, 7 d: Colonies deep, sulcate; margins slightly irregular; mycelium white; texture floccose; sporulation sparse; soluble pigments absent; exudates absent; reverse primrose (66). DG18 25 °C, 7 d: Colonies moderately deep, plane; margins entire; mycelium white; texture floccose; sporulation sparse; soluble pigments absent; exudates absent; reverse buff (45). OA 25 °C, 7 d: Colonies low, plane; margins entire; mycelium white; texture floccose; sporulation moderately dense, dark green (21); soluble pigments absent; exudates absent; reverse white. CREA 25 °C, 7 d: Acid production absent.

Ascomata not observed. Hülle cells hyaline, globose to ovoid, 11–18 μm wide. Conidiophores with smooth stipes, hyaline to light brown, 200–500 × 4–5.5 μm; vesicles hyaline, subclavate, 10–17 μm wide, fertile over the upper two thirds, metulae hyaline, 7–8.5 × 3.5–4.5 μm; phialides hyaline, flask-shaped, 6–8.5 × 2.5–3.5 μm. Conidia echinulate, green in mass, globose, 3–4 μm.

Notes: *Aspergillus sichuanensis* is phylogenetically close to *A. heyangensis*, a species discovered and named after HeYang county, ShanXi province, China. *Aspergillus sichuanensis* can be differentiated from *A. heyangensis* by ITS (96% similarity, 421/435 bp), *BenA* (91% similarity, 378/414 bp), *CaM* (89% similarity, 409/458 bp) and *RPB2* (94% similarity, 713/757 bp) sequences. The distinguishing morphological features of *A. heyangensis* are its brown conidia in mass and lack of Hülle cells [67]. The echinulate, green conidia resemble those of *A. crustosus*, but in *A. crustosus* the vesicles are narrower, (5.5)–7–10–(12) μm, and the conidiophores are shorter (150–250 μm) [17] (Table 2).

*Aspergillus tibetensis* B.D. Sun, X.Z. Jiang & A.J. Chen, sp. nov. Figure 5.

MycoBank: MB837901.

Diagnosis: *Aspergillus tibetensis* grows restrictedly on all media and produces orange pigmented mycelia after one week of incubation.

Typification: CHINA, Tibet, Nyingchi, isolated from rhizosphere soil of willow, 2017, P.P. Huang (holotype HMAS 248371). Ex-type culture CGMCC3.19707 = MN110445. GenBank: ITS = MN640763; *BenA* = MN635249; *CaM* = MN635260; *RPB2* = MN635272.

Etymology: Latin, name refers to its origin, isolated from Tibet, China.

Description: Colony diam—7 d (mm): CYA 13–17; CYA 37 °C No growth; MEA 20–22; OA 19–21; YES 28–31; CREA 14–17; DG18 7–9. Colony characters—CYA 25 °C, 7 d: Colonies low, plane; margins entire; mycelium white; texture floccose; sporulation sparse; conidia en masse pale green (19); soluble pigments absent; exudates absent; reverse buff (45). MEA 25 °C, 7 d: Colonies moderately deep, plane; margins entire; mycelium white; texture floccose; sporulation sparse; conidia en masse pale green (19); soluble pigments absent; exudates absent; reverse buff (45). YES 25 °C, 7 d: Colonies deep, plane; margins entire; mycelium white; texture floccose; sporulation sparse; soluble pigments absent; exudates absent; reverse honey (11) in the middle, buff (45) at edge. DG18 25 °C, 7 d: Colonies moderately deep, plane; margins entire; mycelium white; texture floccose; sporulation absent; soluble pigments absent; exudates absent; reverse buff (45). OA 25 °C, 7 d: Colonies low, plane; margins entire; mycelium white; texture floccose; sporulation absent; soluble pigments absent; exudates absent; reverse pale luteous in the middle, buff (45) at edge. CREA 25 °C, 7 d: Acid production absent.

Ascomata not observed. Hülle cells hyaline, globose to ovoid, 20–26 μm wide. Hyphae hyaline when young, orange pigmented after one wk. Conidiophores with smooth stipes, hyaline, 125–300 × 3.5–4.5 μm; vesicles hyaline, subclavate, 7–12 μm wide, fertile over the upper half to two thirds, metulae hyaline, 2.5–3.5 × 1.5–2.5 μm; phialides hyaline, flask-shaped, 3.5–4.5 × 1.5–2 μm. Conidia smooth, pale green in mass, globose, 2.5–3 μm.

Notes: *Aspergillus tibetensis* grows restrictedly, and its hyphae develop orange pigment after one week, features that differentiate it from other species of section *Aenei*. According to BLAST analyses, ITS of *A. tibetensis* is close to *A. crustosus* (97% similarity), *BenA*, *CaM* and *RPB2* sequences of *A. tibetensis* are close to *A. spectabilis* (94%, 86%, 97% respectively).

## 4. Discussion

In this study, soil samples were collected from 11 provinces and the Tibet autonomous region and ranged across temperate and tropical regions, high and low altitude areas of China. After sorting the isolates into morphological types, preliminary identification revealed 445 *Aspergillus* strains of 6840 fungal strains. These *Aspergillus* species belonged to six subgenera, and among them 93 isolates belonged to subgenus *Nidulantes*, and seven isolates were identified as four new specie belonging to sections *Aenei*, *Ochraceorosei,* and *Sparsi* of subgenus *Nidulantes*, respectively (Appendix A).

Some of the species in subgenus *Nidulantes* produce a sexual state. These species with emericella-like ascomata mostly belong to section *Nidulantes* (*A. stellatus*, *A. aurantiobrunneus*, *A. nidulans* and *A. multicolor* clades) [4] and section *Aenei* (*A. bicolor*, *A. discophorus*, *A. foeniculicola* and *A. spectabilis*) [21,28,68]. Two new species (*A. sichuanensis* and *A. tibetensis*) assigned to section *Aenei* in this study only produced Hülle cells in culture, but not ascospores. In section *Aenei*, all species produce subglobose to subclavate, biseriate conidial heads, however, the length of conidiophores, size of vesicles and conidial ornamentation can be used to distinguish species in this section [17,28].

Species in sections *Sparsi* and *Ochraceorosei* generally originate from tropical or subtropical soils [17,36,37,38,69,70,71]. Two new species (*A. guangdongensis* and *A. guangxiensis*) reported in this study were isolated from adjacent subtropical provinces (Guangdong and Guangxi provinces). Section *Sparsi* species are characterized by thin, sparse and submerged mycelia [17], such as observed in *A. guangxiensis*. Other features common in this section include globose to subglobose, biseriate conidial heads that are fertile over the entire surface. Most species in section *Sparsi* produce smooth to finely roughened conidia. The only exception is *A. biplanus*, which produces conspicuously echinulate conidia [17].

Species assigned to section *Ochraceorosei* clustered in two main branches, *A. guangdongensis* clustered together with *A. funiculosus*, while the other two species, *A. ochraceoroseus* and *A. rambellii*, clustered together. The classification of *A. funiculosus* in a species group was originally doubtful, even though Raper and Fennell (1965) accepted *A. funiculosus* as the only uniseriate species in section *Sparsi* (*A. sparsus* group) [17]. Phylogenetic classifications showed that *A. funiculosus* belonged to section *Ochraceorosei* even though bootstrap and Bayesian support values were low [4,34]. Introducing a second uniseriate species, *A. guangdongensis*, from this study, has clarified and stabilized the phylogenetic position of *A. funiculosus*. The two species form a distinct pair of sister species both with uniseriate vesicles. *Aspergillus guangdongensis* produces microcolonies on CYA at 37 °C, however, the growth characteristics of *A. funiculosus* at 37 °C were not mentioned in its original description [72]. Shumi et al. (2004) reported that profuse aerial mycelium was present at 37 °C during an enzyme screening experiment [73]. Not all species in section *Ochraceorosei* species produce Hülle cells. *Aspergillus ochraceoroseus* and *A. rambellii*, which appeared as sister species on a separate branch (Figure 1), are biseriate and do not grow at 37 °C [36,37].

To date, thirty-six species of subgenus *Nidulantes* have been recorded in China. The China General Microbiological Culture Collection Centre (CGMCC) maintains 11 species [4,74,75] from 15 provinces, and most of them originated from soil, air, and moldy materials. The first extensive study of subgenus *Nidulantes* and related species in China [74] was based on morphological identifications and recorded 17 species in subgenus *Nidulantes* and eight species in *Emericella* (assigned in sections *Aenei*, *Nidulantes*, *Covernicolarum* according to current classification). Sun and Qi described *Aspergillus heyangensis* [67], Li et al. (1998) collected and examined 1402 soil samples from northeastern (Jiamusi and Harbin), northern (Beijing), and northwestern (Yunchuan and Xian) China, and recorded eight species and one variety: *Emericella acristata*, *E. corrugate*, *E. foeniculicola*, *E. miyajii*, *E. nidulans* var. *lata*, *E. quadrilineata*, *E. rugulosa* and *E. undulata* [76]. Later, *Aspergillus keveioides* and *A. sigarelli* in section *Usti* and *Emericella miraensis* in section *Nidulantes* were newly described [33,77,78]. With four new species described in this study, subgenus *Nidulantes* is now extended to 40 species in China and includes species in sections *Sparsi* and *Ochraceorosei*. The findings of this study will improve our understanding of the distribution of *Aspergillus* species in China and provide a basis for the further development and application of species in the subgenus *Nidulantes*.

## Figures and Tables

**Figure 1 jof-08-01205-f001:**
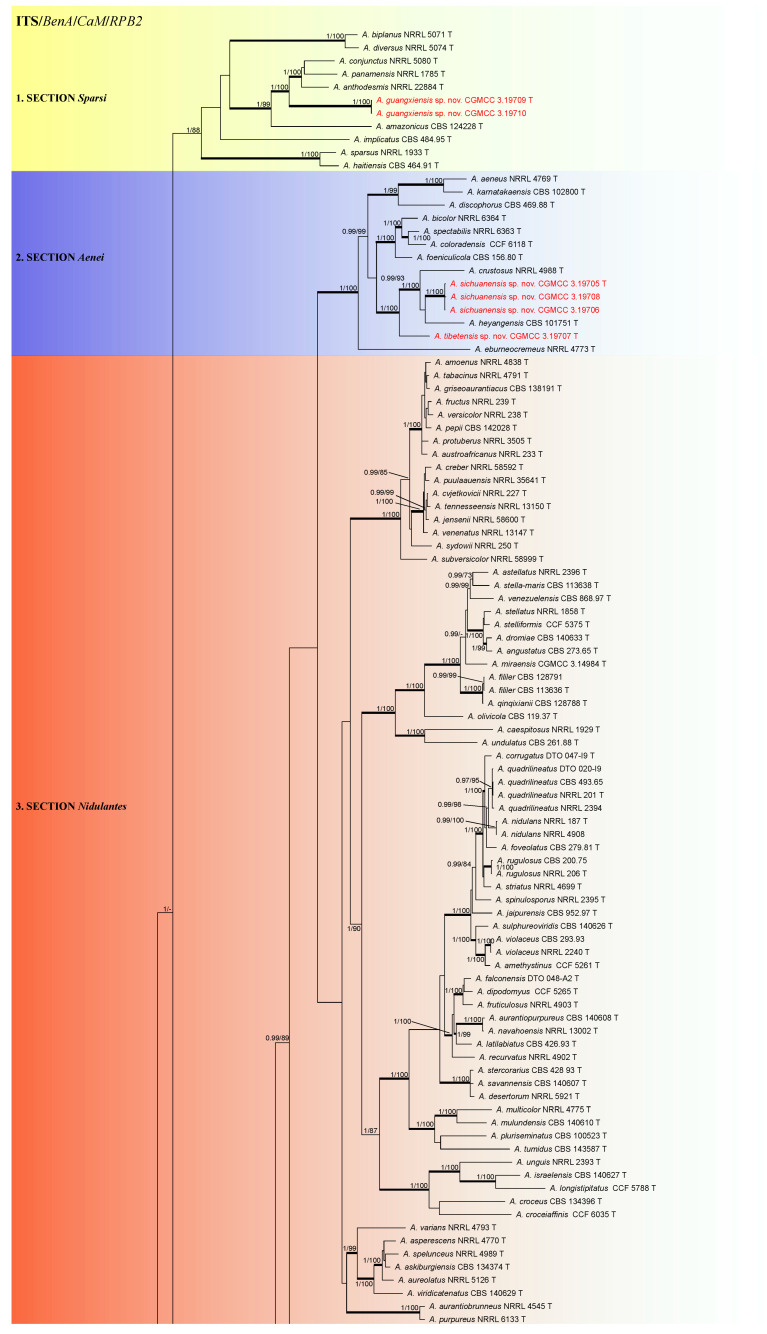
Phylogenetic tree of subgenus *Nidulantes* inferred from concatenated four loci: ITS, *BenA*, *CaM* and *RPB2*. Branches with values more than 1.00 pp and 95% bs are thickened. The phylogram is rooted with *Aspergillus flavipes* NRRL 302 T. T = ex-type.

**Figure 2 jof-08-01205-f002:**
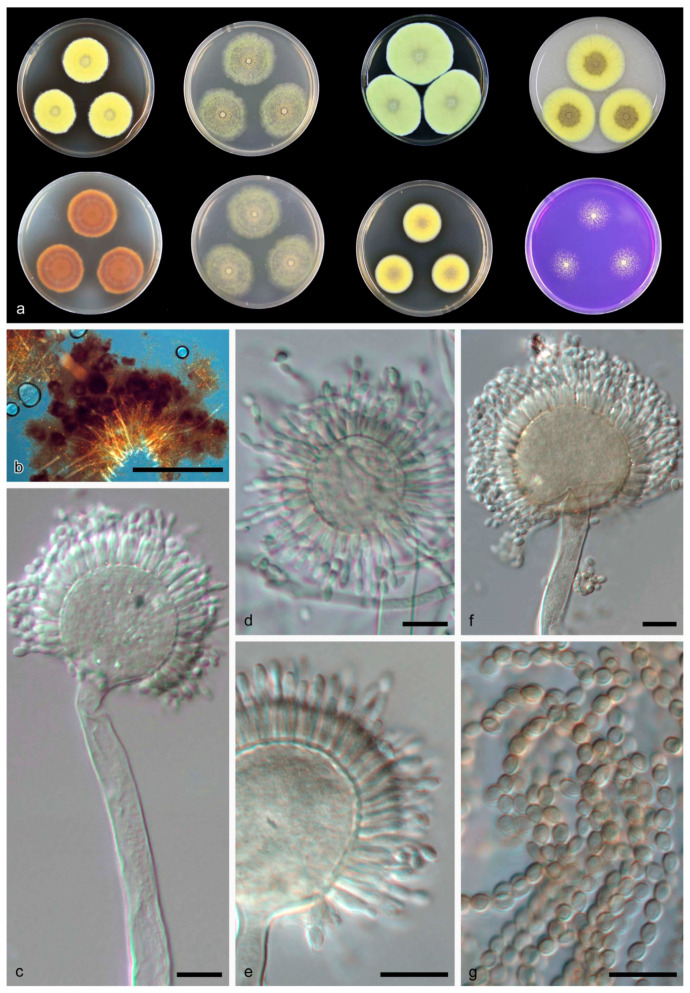
*Aspergillus guangdongensis* CGMCC 3.19704 T. (**a**) Colonies: top row left to right, obverse CYA, obverse MEA, YES and OA; bottom row left to right, reverse CYA, reverse MEA, DG18 and CREA. (**b**–**f**) Conidiophores. (**g**) Conidia. Scale bars: (**b**) = 500 μm; (**c**–**g**) = 10 μm.

**Figure 3 jof-08-01205-f003:**
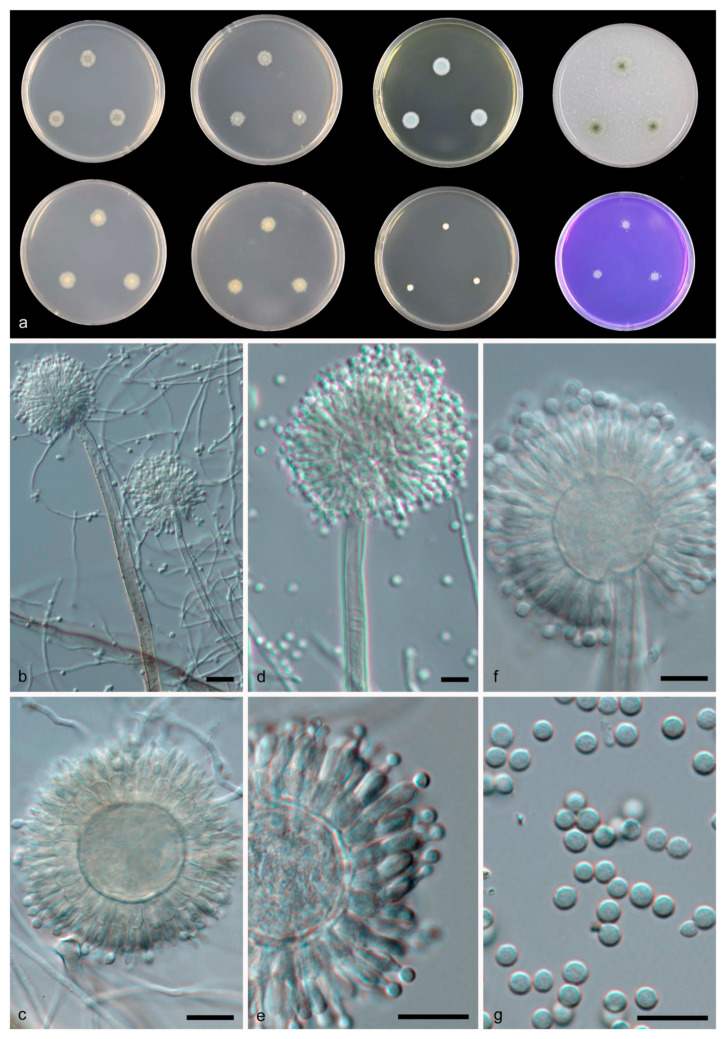
*Aspergillus guangxiensis* CGMCC 3.19709 T. (**a**) Colonies: top row left to right, obverse CYA, obverse MEA, YES and OA; bottom row left to right, reverse CYA, reverse MEA, DG18 and CREA. (**b**–**f**) Conidiophores. (**g**) Conidia. Scale bars: (**b**) = 20 μm; (**c**–**g**) = 10 μm.

**Figure 4 jof-08-01205-f004:**
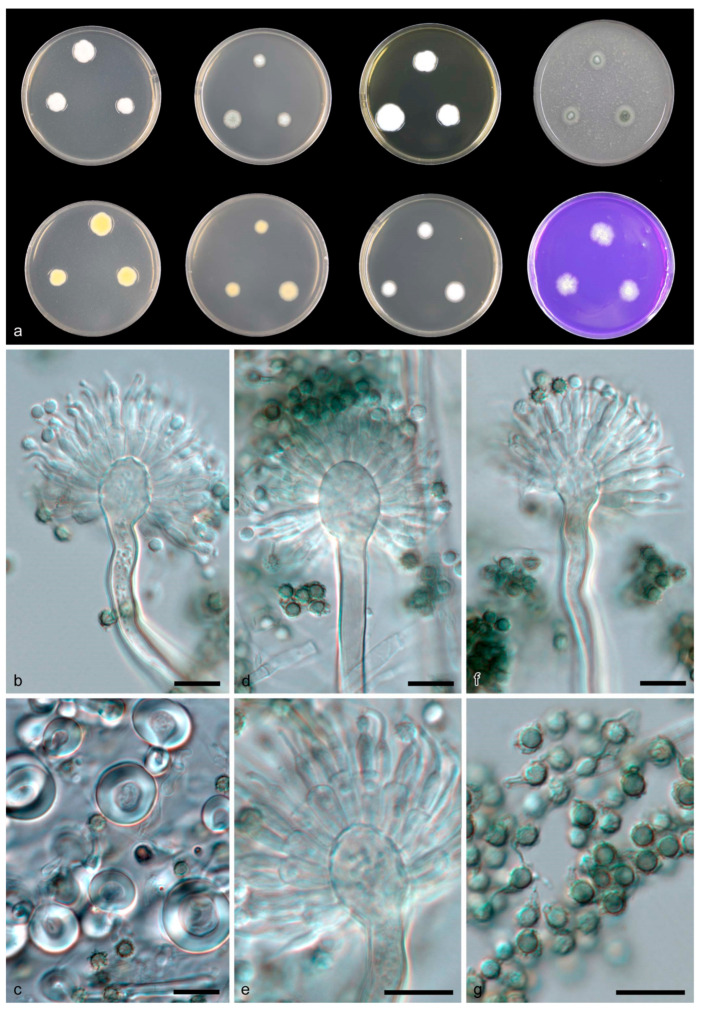
*Aspergillus sichuanensis* CGMCC 3.19705 T. (**a**) Colonies: top row left to right, obverse CYA, obverse MEA, YES and OA; bottom row left to right, reverse CYA, reverse MEA, DG18 and CREA. (**b**,**d**–**f**) Conidiophores. (**c**) Hülle cells. (**g**) Conidia. Scale bars: (**b**–**g**) = 10 μm.

**Figure 5 jof-08-01205-f005:**
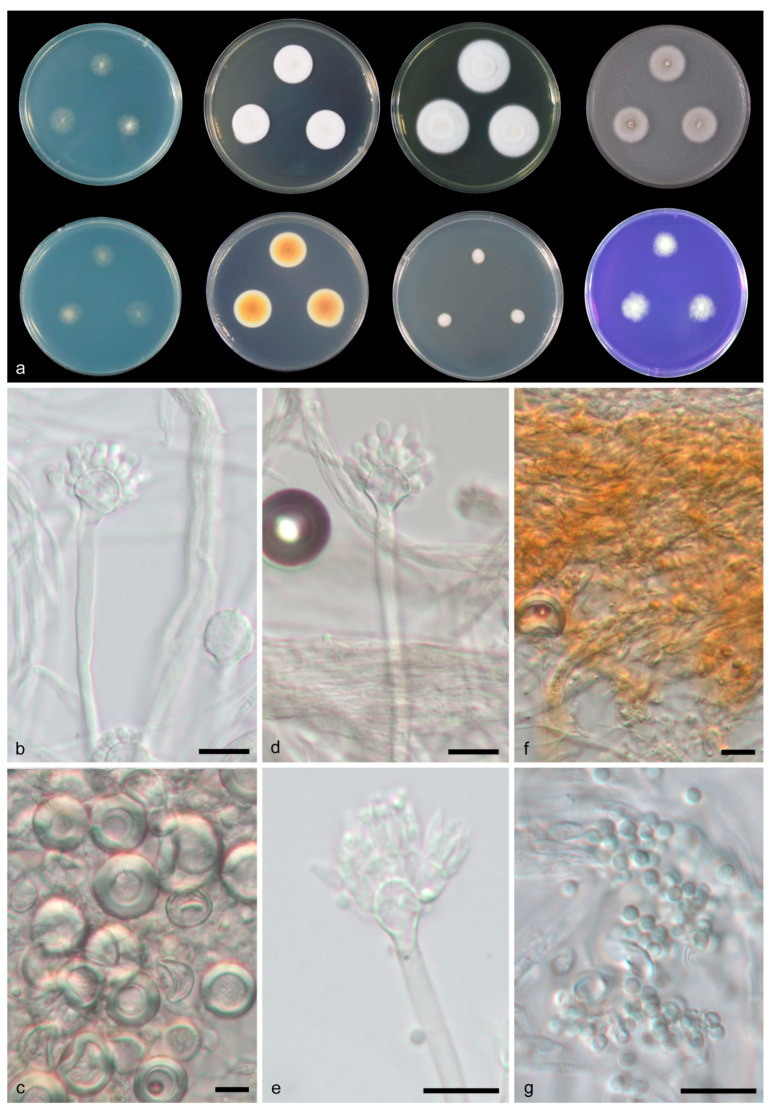
*Aspergillus tibetensis* CGMCC 3.19707 T. (**a**) Colonies: top row left to right, obverse CYA, obverse MEA, YES and OA; bottom row left to right, reverse CYA, reverse MEA, DG18 and CREA. (**b**,**d**–**f**) Conidiophores. (**c**) Hülle cells. (**g**) Conidia. Scale bars: (**b**–**g**) = 10 μm.

**Table 1 jof-08-01205-t001:** Strains used in this study.

Species	Origin	Strain No.	ITS	*BenA*	*CaM*	*RPB2*
*Aspergillus guangdongensis*	Farmland soil, Guangdong	CGMCC 3.19704 T	MN640760	MN635246	MN635257	MN635269
*A. guangxiensis*	Farmland soil, Guangxi	CGMCC 3.19709 T	MN640765	MN635251	MN635262	MN635274
*A. guangxiensis*	Farmland soil, Guangxi	CGMCC 3.19710	MN640766	MN635252	MN635263	MN635275
*A. sichuanensis*	Farmland soil, Sichuan	CGMCC 3.19705 T	MN640761	MN635247	MN635258	MN635270
*A. sichuanensis*	Farmland soil, Sichuan	CGMCC 3.19706	MN640762	MN635248	MN635259	MN635271
*A. sichuanensis*	Farmland soil, Sichuan	CGMCC 3.19708	MN640764	MN635250	MN635261	MN635273
*A. tibetensis*	Farmland soil, Tibet	CGMCC 3.19707 T	MN640763	MN635249	MN635260	MN635272

**Table 2 jof-08-01205-t002:** Morphological comparisions of new species and their closely related species.

	Macromorphology Colony Diam 25 °C, 7 d (mm)			Micromorphology (μm)							
Species	CYA	CYA 37 °C	MEA	Conidial Head	Vesicle	Stipe Length	Conidia Ornamentation	Conidia Shape and Size	Ascomata	Hülle Cells	References
* **Aspergillus guangdongensis** *	30–33	3–4	43–44	Uniseriate	25–35	230–450	Smooth	Subglobose to ellipsoidal, 3–4 × 2–3	NOB	NOB	This study
*A. funiculosus*	* 30–35 after 10–14 d on CA	-	-	Uniseriate	8–35	400–600	Smooth to coarsely rugulose	Elliptical to globose, 3–3.5 × 2–2.5	NOB	NOB	[17]
* **A. guangxiensis** *	11–13	NG	9–11	Biseriate	17–28	350–600	Finely roughened	Subglobose to globose, 3–4	NOB	NOB	This study
*A. conjunctus*	* 30 after 10 d on CA	-	40 after 10 d	Biseriate	15–30	500–700	Smooth to finely roughened	Subglobose to globose, 2.5–4	NOB	elongate	[17]
*A. haitiensis*	30–35 after 14 d	-	50–60 after 14 d	Biseriate	10–25	200–500	Smooth	Globose to ellipsoidal, 4–5.6 × 5–6	NOB	NOB	[38]
* **A. sichuanensis** *	14–23	5–6	13–16	Biseriate	10–17	200–500	Echinulate	Globose, 3–4	NOB	globose to ovoid	This study
*A. heyangensis*	19–23	NG	-	Biseriate	8–12	60–150–(200)	Roughened	Subglobose to globose, 2.5–3.5	NOB	NOB	[67]
*A. crustosus*	* 20–25 after 14 d on CA	-	-	Biseriate	(5.5)–7–10– (12)	150–250	Echinulate	Granular to echinulate, 2.5–3	NOB	globose to irregular	[17]
* **A. tibetensis** *	13–17	NG	20–22	Biseriate	20–26	125–300	Smooth	Globose, 2.5–3	NOB	globose to ovoid	This study

* CA = Czapek’ solution agar; NG = No growth; NOB = not observed.

## Data Availability

The alignment sequences and the tree file were deposited in TreeBASE (S29781, https://www.treebase.org, accessed on 14 October 2022).

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
