# Peer review of "Four New Species of Aspergillus Subgenus Nidulantes from China"

_jof, 2022, doi:10.3390/jof8111205_

Round 1
Reviewer 1 Report
This taxonomic study proposes 4 new species of Aspergillus and uses common tools for describing species. However, there are some methodological ambiguities and stylistic flaws that should be resolved.
I) Text and style: In general, the paper would benefit from English proofreading and the readability and attractivity of the text could also be improved. Sometimes the sentences sound very trivial, simple, and without clear flow and connection between them. The tenses sometimes also alternate strangely in the sentences that follow each other. And there are quite many typos.
II) Abstract
a) there is no information, on why this study is important or why the newly described are important.
b) Some sentences are not important at all for abstract – e.g. „Some of these sections only include a few species, e.g., sections Aeni. „ In addition, in my opinion, sect. Aenei includes quite many species.
c) the authors mentioned that they isolated 6840 isolates but these were not Aspergillus species. I think that the information about a number of Aspergillus strains is more important than the overall number of fungal strains.
d) Here and elswhere, correct Aeni to Aenei
e) you mentioned BLAST analyses but no information is in the text. Include it for every new species.
III) INTRODUCTION
a) tumidu – correct to tumidus
b) A. delacroxii is an illegitimate name, it has to be replaced by A. spinulosporus here and in the tree
METHODS and RESULTS
Where is the information about 6840 isolates and the number of Aspergillus species? Why it is presented as a part of the discussion?
Where is Table S1?
a) Silvati - correct to „Silvatici“ according to Houbraken et al. (2020) here and elsewhere and in the Figures
b) Raperi - correct to „Raperorum“ according to Houbraken et al. (2020) here and elsewhere and in the Figures
c) crustotsus - crustosus
TAXONOMY
a) What is HMAS? Explain in methods.
b) Deposit ex-type isolates into at least one culture collection outside China.
c) What is MN? Explain.
d) What is CSA medium? How can you compare this medium with your data when you haven't used it for cultivation? I don't think that these comparisons are relevant and other features should be included in the diagnoses.
DISCUSSION
a) The first paragraph includes results.
b) “ornamentation can be used to distinguish species” – ornamentation of which structure?
c) “biseriate vesicles” – this is not correct, not vesicles but conidiophores or conidial heads can be designated as uni- or biseriate
d) section Ochraceoroseus - Ochraceorosei
4) the last paragraph is very chaotic and includes too diverse information. Rephrase.
Author Response
- I) Text and style: In general, the paper would benefit from English proofreading and the readability and attractivity of the text could also be improved. Sometimes the sentences sound very trivial, simple, and without clear flow and connection between them. The tenses sometimes also alternate strangely in the sentences that follow each other. And there are quite many typos.
Reply: The text was revised accordingly.
- II) Abstract
- a) there is no information, on why this study is important or why the newly described are important.
Reply: The species number of subgenus Nidulantes grows rapidly recently, however large-scale sample and research is absent in China. This information was added in L30. Furthermore, the importance of this study were detailly described in discussion part L382- L385.
- b) Some sentences are not important at all for abstract – e.g. „Some of these sections only include a few species, e.g., sections Aeni. „ In addition, in my opinion, sect. Aenei includes quite many species.
Reply: This sentence was deleted.
- c) the authors mentioned that they isolated 6840 isolates but these were not Aspergillus species. I think that the information about a number of Aspergillus strains is more important than the overall number of fungal strains.
Reply: Information of Aspergillus strains was added to replace the overall fungal strains number.
- d) Here and elswhere, correct Aeni to Aenei
Reply: Aeni was corrected as Aenei in text.
- e) you mentioned BLAST analyses but no information is in the text. Include it for every new species.
Reply: BLAST analyses results were added in text.
III) INTRODUCTION
- a) tumidu – correct to tumidus
Reply: Done
- b) A. delacroxii is an illegitimate name, it has to be replaced by A. spinulosporus here and in the tree
Reply: Done
METHODS and RESULTS
Where is the information about 6840 isolates and the number of Aspergillus species? Why it is presented as a part of the discussion?
Where is Table S1?
Reply: It was submitted to system.
- a) Silvati - correct to „Silvatici“ according to Houbraken et al. (2020) here and elsewhere and in the Figures
Reply: Done
- b) Raperi - correct to „Raperorum“ according to Houbraken et al. (2020) here and elsewhere and in the Figures
Reply: Done
- c) crustotsus - crustosus
Reply: Done
TAXONOMY
- a) What is HMAS? Explain in methods.
Reply: Done
- b) Deposit ex-type isolates into at least one culture collection outside China.
Reply: Depositing strains abroad is restricted at this time.
- c) What is MN? Explain.
Reply: MN is the strain code in Moon (Guangzhou) Biotech Ltd
- d) What is CSA medium? How can you compare this medium with your data when you haven't used it for cultivation? I don't think that these comparisons are relevant and other features should be included in the diagnoses.
Reply: This sentence is revised
DISCUSSION
- a) The first paragraph includes results.
Reply: This first paragraph summarizes all the results for later discussion.
- b) “ornamentation can be used to distinguish species” – ornamentation of which structure?
Reply: “ornamentation can be used to distinguish species” was revised as “conidial ornamentation can be used to distinguish species”.
- c) “biseriate vesicles” – this is not correct, not vesicles but conidiophores or conidial heads can be designated as uni- or biseriate
Reply: “biseriate vesicles” was revised as “biseriate conidial heads”.
- d) section Ochraceoroseus - Ochraceorosei
Reply: Done.
4) the last paragraph is very chaotic and includes too diverse information. Rephrase.
Reply: Done.
Reviewer 2 Report
Dear Authors,
This paper describes four new species of Aspergillus subgenus Nidulantes based on morphological and molecular data. The descriptions and methods are both well-written. However, there are a number of issues that must be addressed and corrected.
Concerning two major issues (suggestions):
- The morphological data seems to be convincing that these are four novel species, and the phylogenetic analysis is adequate to support the species classification, but I think the phylogenetic tree is hard to see! So, could you please replace it or improve it?
- Could you please add one more table, as suggested in Table 2, so that the reader can easily compare the four novel taxa? "Morphological comparisons of new species and closely related species in DISCUSSION based on some/selected synthetic media (CYA MEA, YES; CREA and DG18) under same condition".
Minor corrections to the pdf file (please see attached file)
- What is the number assigned to the tree in TreeBase?
- Some suggest changing the words or highlighting certain words in orange and red.
- Please provide some citations to support the content in the MS.
- An abbreviation in loci molecular names is a shortened form of a written word or phrase. Abbreviations can be used to save space by eliminating the need to repeat long words and phrases. Some loci are mentioned in the INTRODUCTIOIN PART with the full name and immediately follow it with the abbreviated version.

Author Response
1.The morphological data seems to be convincing that these are four novel species, and the phylogenetic analysis is adequate to support the species classification, but I think the phylogenetic tree is hard to see! So, could you please replace it or improve it?
Reply: Done.
2.Could you please add one more table, as suggested in Table 2, so that the reader can easily compare the four novel taxa? "Morphological comparisons of new species and closely related species in DISCUSSION based on some/selected synthetic media (CYA MEA, YES; CREA and DG18) under same condition".
Reply: Done.
Minor corrections to the pdf file (please see attached file)
- What is the number assigned to the tree in TreeBase?
Reply: TreeBase number was added
- Some suggest changing the words or highlighting certain words in orange and red.
Reply: Done.
- Please provide some citationsto support the content in the MS.
Reply: Done.
- An abbreviation in loci molecular names is a shortened form of a written word or phrase. Abbreviations can be used to save space by eliminating the need to repeat long words and phrases. Some loci are mentioned in the INTRODUCTIOIN PART with the full name and immediately follow it with the abbreviated version.
Reply: Done.
Reviewer 3 Report
Dear Authors,
This is a good presentation of descriptions of several new taxa. Most of my suggestions are editorial and can be found in the attached file.
Best wish

Author Response
Reply: the text was revised according to suggestions noted in attachment.
Round 2
Reviewer 1 Report
1) in my opinion, English was not improved
2) diagnosis of Aspergillus guangxiensis (lines 255-256) - my previous comment was not addressed. What is CA medium? How can you compare this medium with your data when you haven't used it for cultivation? I don't think that these comparisons are relevant and other features should be included in the diagnoses.
3) diagnosis of Aspergillus tibetensis - you stated that "it grows restrictedly, and its hyphae develop orange pigment after one week, features that differentiate it from other species of section Aenei." But where are some data from other species? What is this subjective claim based on?
4) It is hard to believe that strain cannot be deposited elswhere outside China. How can you ensure the accessible of these strains to the scientific community?
Author Response
Review-1
- in my opinion, English was not improved
Response: Revisions were made in L15, L21, L31, L51, L58, L60, L71-73, L96, L99-100, L111, L118, L133-134, L139, L159, L300, L362, L383-386. Professor Gerald Bills, one of co-authors has improved the English writing part.
2) diagnosis of Aspergillus guangxiensis (lines 255-256) - my previous comment was not addressed. What is CA medium? How can you compare this medium with your data when you haven't used it for cultivation? I don't think that these comparisons are relevant and other features should be included in the diagnoses.
Response: CA medium is Czapek' solution agar, this was clarified in Table 2. This medium is widely used previously (20 years ago) to define the growth ability (restricted growth, mediate growth, or fast growth). At present CYA medium is more widely used instead of CA medium in Aspergillus species, the key component of CYA medium is Czapek' solution, similar to CA medium. The only difference is the addition of Yeast extract in CYA medium, which do not affect the growth rate of Aspergillus species significantly, Aspergillus strains shows similar growth patten on CYA and CA. To better compare our new species on CA related medium (CA, CYA), we prefer to keep this important diagnosis.
3) diagnosis of Aspergillus tibetensis - you stated that "it grows restrictedly, and its hyphae develop orange pigment after one week, features that differentiate it from other species of section Aenei." But where are some data from other species? What is this subjective claim based on?
Response: In our previous work on Subgenus Nidulantes (Chen et al. 2016; Sun et al. 2017, we have studied most of type species in this subgenus, and we have never observed these phenotype in other section Aenei species.
References:
- Chen, A.J.; Frisvad, J.C.; Sun, B.D.; Varga, J.; Kocsubé, S.; Dijksterhuis, J.; Kim, D.H.; Hong, S.B., Houbraken, J.; Samson, R.A. Aspergillus section Nidulantes (formerly Emericella): Polyphasic taxonomy, chemistry and biology. Stud Mycol 2016, 84, 1–118.
- Sun, B.D.; Ding, G.; Zhang, Y.S.; Zhao, G.Z.; Zhou, Y.G.; Chen, A.J. Current taxonomy of Aspergillus subgenus Nidulantes and re-identification of several strains. Mycosystema 2017, 36, 1192–1209.
4) It is hard to believe that strain cannot be deposited elswhere outside China. How can you ensure the accessible of these strains to the scientific community?
Response: We have tried several times, but for the time being it’s hard to do this. Other researchers can obtain these strains via CGMCC.
Reviewer 3 Report
Dear Authors,
Overall nice job on this revised version.
Thanks
Author Response
Thanks for the comments.